# REFINING SPECS FOR LLM-BASED RTL AGILE DESIGN

## ABSTRACT

Large language models (LLMs) are increasingly employed to assist in agile register-transfer-level (RTL) hardware design. This is a labor-intensive stage in developing FPGA-based acceleration services or prototyping ASICs, and successful automation can largely shorten the development cycle. However, benchmarks are reporting a relatively low functional correctness rate (sometimes called accuracy) when generating simple modules of less than 100 lines of code (LOC, in Verilog), questioning the practicality of current LLMs for real-world designs. This paper highlights that the low accuracy is attributed to the use of low-quality descriptions as prompts in both training datasets and benchmarks. First, the natural language descriptions (NLDs) do not contain all the semantics constrained by the testbenches (TB), causing false negatives during verification. Second, existing automatically generated NLDs are usually too detailed in implementation, which is not suitable for both training and benchmarking. We designed tools to quantify the clarity and simplicity of the cases, improve the quality of existing and future LLM-for-RTL datasets, and assist agile RTL designers in creating qualified specifications (specs, i.e., formatted and complete NLDs). We show by experiment that LLMs can create specs with high quality at a low cost. Additionally, when equipped with these specs, general-purpose LLMs can achieve a high pass@5 rate (up to 89% on RTLLM, 96% on VerilogEval-Human) without requiring expensive fine-tuning or post-generation self-fixing.

## 1 INTRODUCTION

In the recent past, large language models (LLMs) have emerged as general-purpose design assistants, and their capabilities have been examined in various aspects of software design, including code generation, debugging, and refactoring (Schmid et al., 2025; Esposito et al., 2025). Following this idea, researchers have begun to explore their potential in automatic, agile hardware design, particularly in register-transfer-level (RTL) design and verification, which facilitates the rapid development of FPGA accelerators or ASIC prototypes. The use case is usually defined as follows. The user prepares a natural language description (NLD) of the to-be-designed hardware, and a verification method such as an executable testbench (TB) or a formal verifier (FV). The LLMs are instructed to generate an RTL design according to the NLD, referred to as the device under test (DUT), that should pass the verification over the TB or the FV.

Benchmarks have been proposed to evaluate the quality of designs, especially the probability of correctness (Lu et al., 2024; Liu et al., 2023; Jin et al., 2025; Purini et al., 2025). Unfortunately, the results are not satisfactory. According to the OpenLLM paper (Liu et al., 2024), GPT-4, the leading general-purpose model of the time, managed only 55.8% pass@5 on the VerilogEval-Human benchmark (Liu et al., 2023), and 65.5% on the RTLLM benchmark (Lu et al., 2024). The tasks in these benchmarks are mostly simple components, so these low pass@5 scores suggest that LLMs are not yet practical for generating real-world designs directly.

Improvements are proposed for both the models and the generation strategies, i.e., the prompts and workflows. Works on model fine-tuning focus on how to automatically collect large-scale datasets and create better NLDs for training (Zhu et al., 2025; Zhang et al., 2024). Code segments from real-world projects are serving as golden models (GMs), and NLDs and TBs are created from them. Meanwhile, works on generation strategies focus on automatically correcting errors after an initial

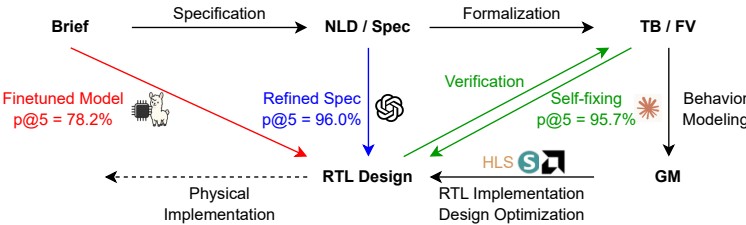

Figure 1: The workflow of traditional and agile RTL design. Black arrows mark the regular design workflow. Red (Li et al., 2025), blue (our work), and green (Zhao et al., 2024b) arrows mark the LLM-based agile design workflow. The percentages are the best reported pass rate on the VerilogEval-Human benchmark.

attempt (Li et al., 2025; Zhao et al., 2024b). Error reports created by simulators are fed back to the LLMs, and a correct design can be sampled within a few iterations. However, these efforts still have shortcomings: the accuracy improvement is limited (especially when fine-tuning from small models), scoring high only on line-by-line translation benchmarks (e.g., VerilogEval-Machine), or consuming several times more computing power.

Interestingly, through manual inspection of existing datasets and benchmarks, we found that many failures do not originate from inherent limitations of current LLMs. Instead, a substantial portion of the failing cases should be reinterpreted as false negatives (FNs), caused by two recurrent deficiencies in the input NLDs: (D1) insufficient or misleading semantic specifications, and (D2) overly complicated implementation-level instructions. Case studies and qualitative analyses are presented in Section 3. After correcting these deficiencies, LLMs are often able to generate functionally correct designs. This raises an important question: to what extent have existing benchmarks underestimated the true power of LLMs in RTL generation?

This paper proposes a paradigm for writing specifications (specs) in a fine format with **clarity** and **simplicity**, to prompt the LLM-based RTL design agents. Similar to the traditional workflow, human designers should provide a precise definition of functionalities, sequential behaviors, and semantics of the parameters, ports, and I/O signals before moving to the implementation phase. Since these documents must be produced (as user guides) regardless of whether LLMs design agents are involved, a minimal extra human workload is introduced. In fact, they can also be generated by LLMs when properly prompted. Given the improved specs, RTL generation accuracy can increase by approximately 10-20 percentage points. For example, GPT-4 achieves pass@5 rates of 89.0%, 96.0%, and 99.5% on RTLLM, VerilogEval-human, and VerilogEval-machine, respectively.

The main contributions of this work are:

- We demonstrated the importance of a clear yet simple spec for the accuracy of RTL generating LLMs. We designed templates and automated tools to create such specs, and improved existing benchmarks by replacing the original NLDs with the specs we created.

- We examined that simply using these specs can help LLMs achieve a score similar to (for weaker models) or much better than (for powerful models) existing works. This even holds when the spec is generated by an LLM other than the one that generates the RTL code.

- We show that enhancing the spec quality can also increase the accuracy of generating complex designs, such as FFT and Conv2d, which were previously reported to be challenging.

## 2 BACKGROUNDS

### 2.1 RTL DESIGN WORKFLOW

As illustrated in Figure 1, the traditional RTL design workflow can be divided into stages, each producing a solid outcome that is validated before proceeding to the next stage. These intermediate outcomes, including the specs, TBs / FVs, and GMs, are not only essential for subsequent stages but also serve as user guides and tools incorporated into the released product.

The brief is a short statement of what the module is designed for, typically appearing in the first paragraph of a user manual. The spec is a human-friendly yet complete semantic description of the design. It should be able to guide a user to use the design as a black box, without awareness of the internal details. The TB and the FV are machine-friendly versions of the spec. A design is functionally correct only if it passes verification. Of course, passing does not always imply correctness. But since orthogonal works have attempted to mitigate these false positives (FPs) by writing better TBs (Jin et al., 2025), we will not further discuss the FPs in our paper. The GM is a straightforward implementation, without performance or cost considerations, but simple enough to guarantee its correctness. The final design should be semantically equivalent to the GM, so that a system built using the GM can run as-is if the GM is replaced by it.

## 2.2 LLM-BASED AGILE DESIGN

LLMs have been adopted to assist human agile designers to leap from each stage directly to the last. Intuitively, the earlier the stage at which the LLMs start, the less human effort is required, but also the less information is provided, and the greater the difficulty for the LLM. Among these, the last stage (RTL implementation) is typically handled by high-level synthesis (HLS) tools, which lie outside the scope of this paper.

**Metrics and Benchmarks**. The accuracy of the designs is typically measured using the pass@$k$ metric (Chen et al., 2021), which indicates the probability of obtaining at least one passing candidate among $k$ random samples. Each test case should be equipped with either a GM (for fuzzing-based differential tests), a TB (for constructed tests), or an FV (for formal verification over constraints) to determine whether a DUT passes. Although earlier benchmarks relied more on simulation-based TBs, FVs are increasingly employed (Fang et al., 2024; Jin et al., 2025).

RTLLM (Liu et al., 2024) and VerilogEval (Liu et al., 2023) are currently the most recognized benchmarks in this scenario. The latest version of RTLLM (-v2) contains 50 basic industrial function units, including arithmetic units, buffers, FSMs, and signal processing units such as edge detectors or parallel-serial converters. VerilogEval contains 156 crafted tasks collected from beginner exercises: implementing very simple logic with fewer than 10 lines of code (LOC), or translating truth tables or state graphs into code. Two versions of NLDs are provided in different styles. The -v2 (and the previous -Human) version has manually written high-level semantic descriptions, while the -Machine version has LLM-generated GM code summaries.

Recently, benchmarks for generating complex designs or large systems have also been proposed. The authors of AutoSilicon (Li et al., 2025) added generation tasks for FFT, I/O ports, and RISC-V CPU, using the same format as RTLLM. ArchXBench (Purini et al., 2025) provides commonly used functional primitives, such as signal filters and cryptographic algorithms, along with their floating-point and pipelined variants. RealBench (Jin et al., 2025) provides structured design tasks with high-quality manual written specs. Until the submission of this article, these harder tasks have never been solved by any LLM design agent.

**The naive approach**. The earliest attempts directly fed the briefs into general-purpose LLMs or coding models trained with software codes. Minimal human effort is required, but the pass rate is low: only 20%-40% of samples are correct on simple tasks (Thakur et al., 2023). While LLMs are becoming larger and more powerful, this score increased only to 50% (Lu et al., 2024). Further improvements were mainly proposed from three perspectives, as also illustrated in Figure 1.

**Dataset construction and model fine-tuning**. Some believe that the low accuracy is due to a lack of training materials for hardware design. So, they build large-scale datasets (Zhu et al., 2025), including cases involving new input formats (e.g., truth tables and state diagrams (Liu et al., 2025a)) or new hardware description languages (HDLs, e.g., the emerging Chisel (Zhao et al., 2024a)). Then, they fine-tune the models, either supervised or via reinforcement learning. These works usually use small backbone models (with ≤32B parameters) because training larger models is too expensive. This category is exemplified by CodeV-R1 (Zhu et al., 2025).

**Pre-generation prompt engineering**. In parallel with fine-tuning, some works turn to teaching the LLMs through prompts. They highlight the difference between HDL and software codes, decompose states or submodules, or instruct the LLMs to do so by themselves (referred to as self-planning in the RTLLM paper (Lu et al., 2024)). This category is usually employed as part of other approaches.

However, if the backbone model is powerful, using it alone can also yield significant performance improvements. Our work falls into this category.

**Post-generation self-fixing**. In addition to improving the single-shot performance, post-generation mechanisms are proposed to fix minor errors reported during synthesis or simulation. Post-generation mechanisms can achieve a high pass rate of 95.7%, but incur significantly higher inference costs. For example, Mage sampled 20 code candidates, selected the best two, and debugged each for 5 iterations (Zhao et al., 2024b). This is much more expensive than sampling 5 independent candidates, which is the practice of a regular pass@5. This category is exemplified by Mage (Zhao et al., 2024b). Models that are fine-tuned with a self-fixing corpus (given erroneous code and error report), such as CraftRTL (Liu et al., 2025a), can be regarded as merging the self-fixing step into generation, and can also be considered in this category.

## 3    THE REFINED SPEC

In this section, we first highlight the two major deficiencies with existing datasets and benchmarks for LLM-based RTL design tasks. Then, we outline a writing paradigm for high-quality specs and present our LLM-based tools that can help dataset creators or hardware designers to improve their specs, either automatically or interactively. We introduce the **clarity** and **simplicity** metrics and use their joint distribution to evaluate the quality of the datasets before and after refinement. Finally, we discuss related works on benchmark improvements and future directions.

### 3.1    CASE STUDY 1: THE NLD-GM MISMATCH

Deficiency (**D1**) is the semantic mismatch between the NLDs and GMs, TBs, or FVs. Consider the `serial2parallel` task from the RTLLM v2 benchmark (Liu et al., 2024), as shown in Figure 2. This module is normally expected to have eight states, each processes one bit, to accept bitstreams continuously. But the GM creates an extra state, in which `cnt == 4'd8`, skipping every ninth bit received. While this feature is not explicitly described in the NLD, the TB requires the DUTs to have strictly consistent behavior with the GM. Therefore, it is almost impossible for the DUTs (whether written by humans or LLMs) to pass this test unless the designer is allowed to infer unwritten requirements from simulation error reports.

> Implement a series-parallel conversion circuit. It receives a serial input signal "din_serial" along with a control signal "din_valid" indicating the validity of the input data. The module operates on the rising edge of the clock signal "clk" and uses a synchronous design. The input din_serial is a single-bit data, and when the module receives 8 input data, the output dout_parallel outputs the 8-bit data (The serial input values are sequentially placed in dout_parallel from the most significant bit to the least significant bit), and the dout_valid is set to 1.

```
always@(posedge clk or negedge rst_n)begin
  // ...
  else if(din_valid)
    cnt <= (cnt == 4'd8)?0:cnt+1'b1;
  else
    cnt <= 0;
  end
always@(posedge clk or negedge rst_n)begin
  // ...
  else if(cnt == 4'd8)begin
    dout_valid <= 1'b1;
    dout_parallel <= din_tmp;
  end
  // ...
end
```

Figure 2:    The original NLD and GM of the RTLLM `serial2parallel` task (Lu et al., 2024). The red font marks the mismatch between the NLD and the GM.

(**D1**) is harmful for both benchmarks and datasets. Mismatches in benchmarks lead to an underestimation of LLMs' ability. What is worse, mismatches in datasets should be regarded as a form of mislabeling, which, in theory, reduces the accuracy of the trained model.

(**D1**) occurs in both manually-written NLDs and LLM-generated summaries. Recall that in the traditional workflow, a human designer must first present the NLD, then formalize it as a TB or FV, and finally present a semantically equivalent GM. During this process, constraints are applied to downstream outcomes, but upstream documents are not always updated accordingly. Thus, mismatches occur. On the other hand, existing works that automatically create large-scale datasets often prompt the LLMs to "summarize" the functions of modules to generate NLDs. However, the generated summaries may omit the semantic details that are supposed to be in the context. In other words, the LLMs think they are writing an introduction instead of a complete spec.

As the MG-Verilog paper concluded, "although high-level global summaries are the most user-friendly data format, their ambiguity often results in a lack of detailed information necessary for precise code generation" (Zhang et al., 2024). However, their approach of creating multi-grained NLDs leads to another problem, as demonstrated below.

## 3.2 CASE STUDY 2: OVER-SPECIFIED IMPLEMENTATION GUIDES

Deficiency (**D2**) is the over-detailed implementation guidance in automatically generated NLDs. Figure 3 shows the case `#2456` of the MG-Verilog training dataset. This paper proposed a multi-grained training set, claiming that "high-level descriptions can facilitate user-friendly LLM interactions, while detailed descriptions are crucial for enabling LLMs to create complex designs" (Zhang et al., 2024). However, probably to solve (**D1**), the authors prompts the LLMs to "be very specific", and the resulting NLDs turned out to be a line-by-line translation of the GM code without high-level semantics. Another typical example is the VerilogEval-Machine benchmark (Liu et al., 2023). We speculate that the LLMs may have misunderstood this instruction and became specific on *how* (instead of *what*) to design. Also, the excessively fine-grained block partitioning disperses the semantic context, making it difficult for the model to produce meaningful abstractions.

(**D2**) is also harmful for both benchmarks and datasets. Low-level benchmarks cannot assess LLMs' pattern recombination skill because they focus on translating statements rather than selecting them to build functionality. Moreover, in a real-world use case, it is impractical for designers to write low-level NLDs, as this requires the same effort as manually designing the module. On the other hand, although multi-grained curriculum learning is theoretically beneficial, the actual accuracy improvement

Figure 3: The prompt for block-level summary generation, the generated NLD, and the GM code of MG-Verilog training set case `#2456`. The colors show the mapping between code lines and description sentences.

of these fine-tuned models is significant only on low-level benchmarks (Zhao et al., 2024a; Zhang et al., 2024). The excessive, unintended low-level training data may have led to overfitting.

It is worth noting that the RTL design corpus of existing LLMs is small, thus they are unfamiliar with microarchitecture-level instructions. Especially, they tend to confuse registers and wires. For this reason, retaining a small portion of low-level data in the datasets is still essential. We speculate that, after models are properly trained, implementation details will no longer be detrimental.

## 3.3 SPEC EVALUATION, REFINEMENT, AND CREATION

Despite these deficiencies, the need for automated NLD generation remains inevitable as the demand for large-scale benchmarks or training datasets grows. So, we designed three LLM-based tools, namely the *evaluator*, the *converter*, and the *creator*. These tools helped us to identify which aspects of the specification are essential for enabling LLMs to produce correct outputs.

The *evaluator* is prompted to score and criticize specs from a user's perspective. It asks an LLM to measure NLDs using two metrics, clarity and simplicity, defined as follows.

- **Clarity**: Can you learn the module's exact behavior from the spec without correcting any misunderstanding by reading the Verilog code?
- **Simplicity**: Can you learn the module's high-level semantics easily without awareness of its internal implementation?

The *evaluator* scores each metric on a scale of 1 to 5 and provides suggestions for improvement. The prompt template and an example response are available in Appendix A. We ran the *evaluator* on the NLDs of both benchmarks and training datasets. Although these scores are produced by the LLMs and are not rigorous, they still reflect the difficulty level that the LLMs themselves experience during training and evaluation. The distributions of the scores are shown in Figure 4. Intuitively, a low

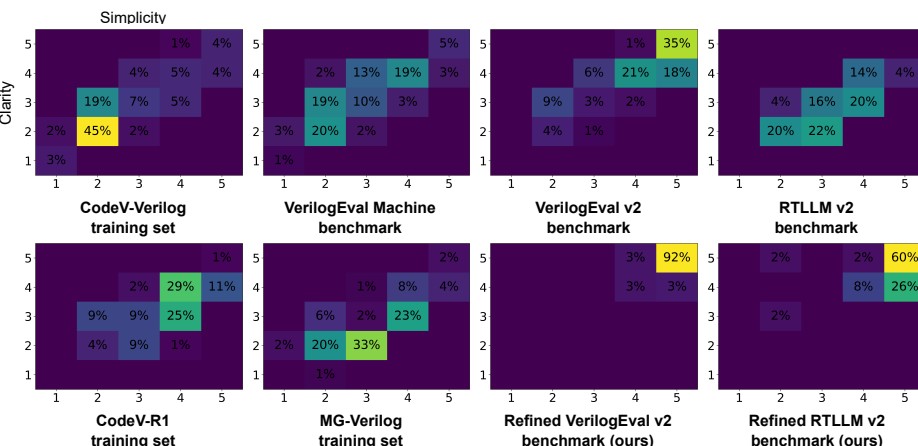

Figure 4: The joint distribution of clarity and simplicity ratings of benchmarks and datasets, scored by our *evaluator* based on GPT-4. 200 cases (or all, if insufficient) are sampled from each dataset.

clarity score means (**D1**), and a low simplicity score means (**D2**). We can see that among the training datasets, the NLDs of MG-Verilog and CodeV-Verilog (both using multi-grained summaries) are of relatively low quality. Meanwhile, the relatively higher quality of the CodeV-R1 dataset may explain why the fine-tuned model performs better.

From the resulting suggestions, we concluded 8 key features to create an ideal spec. First, it should include complete **black-box semantics** to prevent (**D1**), so that if the user replaces the module with another implementation that strictly adheres to the spec, the system should work as-is without any further changes. This includes:

1. **Declaration**. Module name, design signature, and parameters.

2. **Behavior details**. Describes the functional relationship between the output and the input. For arithmetic primitives, this should include the computation formula. When LLMs fail to understand technical terms in a specific area, the spec must explain them.

3. **Sequential features**. Describes reset behaviors, I/O handshaking protocols, and whether the module is pipelined (i.e., can accept another input when processing one). Although intuitively more like implementation details, some FSM states should be regarded as semantics. For example, in a serial-to-parallel module, it is essential to specify whether a frame includes a start bit or parity bits, which introduces extra states.

4. **I/O details**. Describes the semantics and formats of each I/O port and its values, especially control signals (e.g., the function selection signal of an ALU) or complex data structures (e.g., floating point values).

Additionally, in cases where some functionalities are hard to interpret, the spec can include some explanations of the semantics, such as:

5. **Examples** (optional). Describes normal cases and corner cases. The normal cases are examples of how the module will be instantiated and interacted with. The corner cases clarify ambiguous behaviors.

6. **Implementation overview** (optional). Describes the block-level semantics and how blocks collaborate. This follows the idea of many existing works (Zhang et al., 2024; Zhao et al., 2024a; Liu et al., 2025b); however, the blocks should be partitioned by functionality (e.g., FSM states, pipeline stages, or submodules) rather than by syntax (e.g., code lines).

Meanwhile, to prevent (**D2**), restrictions are imposed on the above parts:

7. **Limited RTL details**. The spec should avoid describing the microarchitecture directly; otherwise, the design synthesis task degenerates into a line-by-line translation problem.

This restriction should apply to benchmarks, because real-world designers will not provide these details as mentioned. However, when generating the lowest-level (i.e., RTL) cases for multi-grained datasets, this restriction can be disabled.

8. **Prevent trivial corner case behaviors**. For example, if overflows do not exist or are not expected to be handled, the spec may simply say "assume there are no overflows".

In accordance with this paradigm, we designed the *converter* and the *creator* to refine the specs. We believe this process can be automated by LLMs, because they "excel in summarizing Verilog code rather than generating it" (Zhao et al., 2024a). The prompts used are listed in Appendices B and C.

The *converter* converts existing projects (containing code, documents, tests, etc.) into datasets. It automatically completes or corrects the NLDs using high-level semantics extracted from the GM, the TB, or the FV. Theoretically, when creating datasets without a GM, using a TB or an FV is also fine. However, in the benchmarks we are using, the TBs are too simple for the LLMs to infer the expected behavior. Figure 4 also shows the score distribution of the RTLLM and VerilogEval specs refined by the *converter*. The refined specs have higher scores on both metrics, and we will show in Section 4.2 that they really lead to higher RTL generation accuracy.

The *creator* helps real-world users to create specs for new designs from scratch (i.e., the user cannot provide any of GM, TB, or FV) interactively. It advises possible improvements to the spec based on the comments given by the *evaluator*. Conceptually, the *creator* implements RTLLM's self-planning idea (Lu et al., 2024), guiding the LLMs to reason step by step. We further explicitly instruct it to carefully consider corner cases and sequential behaviors, and to ask users to clarify any ambiguity rather than trust its own hallucinations.

## 3.4 Code Generation

After the specs are created, they can be fed to the LLMs for code generation. Comparing with existing workflows, we replace the NLDs with refined specs and add hints to prevent syntax errors.

As current LLMs still struggle to distinguish between combinational and sequential logic, we add prompts to strengthen LLMs' understanding. We instructed the LLMs to "use `wire` assignments or `always @ (*)` to implement the combinational computations within each clock cycle, and add `_next` suffix to these signal names". This increased the functional correctness rate of FSM tasks, such as RTLLM `radix2_div`.

## 3.5 Discussions of Related Works

Works have proposed other means of NLD refinement or attempted to achieve related goals, such as optimizing the performance, power, and area (PPA), or designing more reliable verification schemes. Here, we discuss some future directions inspired by them.

**NLD clarification**. The NLDs of newer benchmarks, e.g., ArchXBench and RealBench, have better quality but are manually designed (Purini et al., 2025; Jin et al., 2025). We validated our LLM-created specs and found that they have a very similar structure and format to RealBench. While RealBench aims to explore the abilities of the LLMs with harder tasks, we point the way for future benchmark (and training dataset) creators. We believe that our work can overcome "the input format shortcomings in existing benchmarks" (Jin et al. (2025)) with minimal human efforts, while retaining comparability with existing results evaluated on the original version.

**PPA optimization**. RTLLM and AutoSilicon evaluated the PPA of LLM-generated codes (Lu et al., 2024; Li et al., 2025). However, given that the functional correctness rate remains low, we think it is too early to require the LLMs to optimize the PPA of their design. We observed that existing benchmarks include cases that aim to do so but end up being functionally incorrect. An example is given in Section 4.4. Only after removing the PPA optimization statements may the LLMs present test-passing HDL codes "in software style", e.g., using address decoders to access arrays instead of shift registers, although they are more expensive.

Despite this, LLM-generated designs with unsatisfactory PPA can still be used for agile deployment of FPGA-based services, provided their correctness is verified. Accelerators, such as the systolic arrays and spatial accelerators mentioned by some benchmarks (Chang et al., 2024), leverage hard-

ware only for parallelization. A better PPA is usually not the user's main concern if the design can be deployed onto the FPGA without timing or power violations. Meanwhile, trusted computing services may rely on hardware-based security features (e.g., physical isolation (Zhao et al., 2022)) and primitives (e.g., oblivious memory access (Wang et al., 2015)), which need to be deployed on FP-GAs. The sooner the security features are deployed, the lower the risk to the sensitive applications.

**Assertion-based verification and generation**. Most TBs of RTLLM and VerilogEval are based on cycle-accurate difftests (Lu et al., 2024; Liu et al., 2023). So, to reduce FNs, our current *converter* has to require the LLMs to strictly follow the GM. However, removing constraints that are not semantically necessary can also reduce FNs. For example, when designing an accelerator, the microarchitecture and consequently the sequential behavior are not determined until the final implementation stage. So the TB should only verify the correctness of the output value, not the number of cycles required for the outputs to appear.

The authors of RealBench considered the opposite type of error, i.e., the FPs, caused by the low coverage of difftest-based TBs (Jin et al., 2025). They designed FVs for system-level design tasks. In the meantime, AssertLLM (Fang et al., 2024) served as a good step towards using LLMs to translate NLDs into formal SystemVerilog assertions (SVAs). We believe we can further develop this idea by adding assertions to the FV step by step, until it contains sufficiently comprehensive constraints to formally generate the design, e.g., until the logic between each pair of registers is formalized. Because every step can be verified, this approach is theoretically more stable than directly completing the HDL code. We leave this as future work.

## 4 EXPERIMENTS

We conducted experiments to evaluate the refined specs, aiming to answer three questions: (**Q1**) How does the quality of the spec affect the correctness of design generation? (**Q2**) How do the non-basic parts of the spec affect its quality? (**Q3**) Are the better specs helpful in complex tasks?

### 4.1 SETUP

**Benchmarks**. We applied the widely used RTLLM v2, VerilogEval-v2, and VerilogEval-Machine benchmarks (Liu et al., 2024; Pinckney et al., 2024; Liu et al., 2023) to compare with existing works. Additionally, we challenged some more complex tasks from levels 4 and 5 of ArchXBench (Purini et al., 2025). We used the original TBs provided with the benchmarks. We used Synopsys VCS as the simulator, which supports SystemVerilog syntax required by the TBs.

**LLMs**. For spec generation, we chose GPT-4 and Claude-3.5-sonnet (C3.5), the two commonly used general-purpose LLMs. For code generation, we chose Qwen2.5-Coder-32B (QC) as the smaller model, CodeV-R1 (CVR1) as the distilled model, and GPT-4 and C3.5 as the larger models. QC is used as a base model for CodeV (Zhao et al., 2024a) and CVR1 (Zhu et al., 2025), with CVR1 currently the most powerful finetuned model to our knowledge. We fixed the temperature to 0.3.

### 4.2 MAIN RESULTS

**Power of the refined specs**. Table 1 shows the pass@1 and pass@5 scores of LLM-based RTL design agents, given the original NLDs and the full specs (having all eight configurations in Section 3.3 enabled) generated by GPT-4. The specs increased the pass scores of the CVR1, C3.5, and GPT-4 agents on both RTLLM-v2 and VerilogEval-v2 (Human). Remarkably, GPT-4 succeeded on 47 (out of 50) RTLLM tasks and 151 (out of 156) VerilogEval tasks. An in-depth analysis of the failing cases is available in Appendix D, showing that the specs can fix not only the FN cases but also true negative ones (too hard to generate even when the NLD is clear). On VerilogEval-Machine, our spec also helped CVR1 and GPT-4 achieve an almost 100% pass@5 rate.

**Compared with fine-tuning**. We noticed that our specs had a negative effect on QC, especially on the VerilogEval benchmarks. QC is a model trained purely on software codes, so without the implementation guides in the original NLDs, it tends to produce syntax errors. For the CVR1 model, which is also trained from QC, we observe a 16% increase in its pass@5 score. This indicates that the refined specs can work for models powerful enough to interpret them, regardless of their size.

Table 1: The score of LLM-based RTL design agents given the original NLDs and GPT-4 specs, on RTLLM and VerilogEval. Scores in yellow cells are evaluated using RTLLM v1.1 or VerilogEval-Human (v1), which have fewer, easier cases and may yield higher scores. Scores of QC are cited from CodeV-R1 (Zhu et al., 2025). Scores of GPT-4 are cited from CodeV (Zhao et al., 2024a). Other systems marked with (*) are cited from their original papers. SP stands for self-planning. R1D means Distilled R1.

| Agent | Description | | | RTLLM | | VE-H / VE-v2 | | VE-M | |
|---|---|---|---|---|---|---|---|---|---|
| | Base Model | FT Dataset | Strategy | pass@1 | pass@5 | pass@1 | pass@5 | pass@1 | pass@5 |
| CraftRTL-SDG-CC-Repair* | SartCoder2 | Non-textual | Self-fixing | 49.0 | 65.8 | 68.0 | 72.4 | 81.9 | 86.9 |
| QC* | QC-32B | – | – | 47.8 | **63.9** | 47.5 | 60.7 | 66.6 | 76.6 |
| CodeV-All-QC* | QC-7B | Multi-layer | – | – | 55.2 | **56.6** | **67.9** | 81.9 | 89.9 |
| QC-Spec-full (Ours) | QC-32B | – | Spec | **53.6** | 63.1 | 24.7 | 31.3 | 37.1 | 47.8 |
| CVR1* | QC-7B | R1D + Solvable[2] | – | 68.0 | 78.2 | 68.8 | 78.2 | 76.5 | 84.1 |
| CVR1+Spec-full (Ours) | QC-7B | R1D [3] | Spec | **79.6** | **83.7** | **81.8** | **82.0** | **99.6** | **100.0** |
| C3.5-SP | C3.5 | – | SP | 50.0 | 66.1 | 62.4 | 76.8 | – | – |
| MAGE* | C3.5 | – | Self-fixing | – | – | **72.4** | 95.7[4] | – | – |
| C3.5-SP-Spec-full (Ours) | C3.5 | – | SP + spec | **68.4** | **82.1** | 65.8 | 77.4 | – | – |
| GPT-4* | GPT-4 | – | – | – | 65.5 | 43.5 | 55.8 | 60.0 | 70.6 |
| GPT-4-SP[1] | GPT-4 | – | SP | 57.6 | 72.6 | 74.5 | 85.6 | **98.6** | **100.0** |
| GPT-4-SP-Spec-full (Ours) | GPT-4 | – | SP + spec | **77.2** | **89.0** | **84.6** | **96.0** | 93.7 | 99.5 |

**Compared with self-fixing**. On VerilogEval-v2, the state-of-the-art self-fixing design agent (Zhao et al., 2024b) achieves a similar pass@5 score as GPT-4 using our refined spec, while their one-shot pass rate is approximately consistent with earlier works[4]. Although a high score is achieved, this self-fixing process is computationally expensive, as discussed in Section 2.2. While we acknowledge the power of self-fixing, we suggest that a better spec can reduce the required number of iterations.

**Generalizability**. We additionally used C3.5 to generate specs for RTLLM. To prevent a model from embedding information into the specs that only itself can interpret, we performed cross-evaluation: using one model to generate the specs, and another to generate the codes. We measured all the scores with a global temperature of 0.3, so the results may differ from those reported in previous papers.

The scores are shown in Table 2. We can see that both C3.5 and GPT-4 specs can improve the accuracy of distilled and larger models. Among the three NLDs, specs generated by GPT-4 yield the highest pass rates regardless of which model performs code generation. The score of C3.5 using GPT-generated specs is even higher than using specs generated by itself. This shows that one refined-spec version can be beneficial for multiple models.

Table 2: Cross evaluation results on the RTLLM v2 benchmark. All data comes from our experiments.

| | Original NLD | | C3.5 Spec | | GPT-4 Spec | |
|---|---|---|---|---|---|---|
| | pass@1 | pass@5 | pass@1 | pass@5 | pass@1 | pass@5 |
| QC | 44.8 | 52.6 | 43.6 | 50.7 | 53.6 | 63.1 |
| CVR1 | 51.2 | 59.0 | 70.4 | 77.7 | 79.6 | 83.7 |
| CVR1-SP | 59.2 | 73.2 | 56.0 | 66.0 | 72.0 | 82.1 |
| C3.5-SP | 50.0 | 66.1 | 58.8 | 74.2 | 68.4 | 82.1 |
| GPT-4-SP | 57.6 | 72.6 | 66.4 | 79.5 | 77.2 | 89.0 |

We observed a performance decline compared with the original NLD when giving the C3.5 Spec to QC and CVR1-SP (i.e., with self-planning). While we have discussed the limitations of QC, comparing CVR1 with or without SP brings another insight. Although of lower quality, the length of C3.5 specs is similar to GPT-4 specs, making it more difficult for smaller models with limited context length to process. Manual inspection discovered that 14.4% and 15.2% of the CVR1-SP samples have truncated outputs (causing syntax errors) on the original NLD and C3.5, respectively. Without SP, CVR1 achieves a normal pass rate. We conclude that self-planning is unsuitable for smaller models.

**Answer for Q1**. The deficiencies we pointed out are limiting the power of LLM hardware design agents. Given refined specs with high clarity and simplicity, LLMs such as GPT-4 and CVR1 are practical to handle basic RTL design tasks.

---

[1] We implemented our own self-planning prompt for improved accuracy.

[2] *Solvable* means can be solved by a powerful model, which usually implies "no NLD-GM mismatch".

[3] We directly used the released CodeV-R1 model, and no extra training is applied.

[4] MAGE's score is given as pass@1, but this "1 attempt" actually combines 20 samples and 10 debugging iterations. We regard the one-shot correct rate as pass@1, while that after 5 debugging iterations as pass@5.

## 4.3 ABLATION STUDY ON CONFIGURATIONS

We evaluated different configurations of our spec on RTLLM v2. We began with the Spec-basic configuration, which contains features 1-4 as declared in Section 3.3. This forms the essential sections of a spec. Then, we added examples by feature 5 (-cases), and after that, the block-level implementation overview by features 6-8 (-full, where features 7 and 8 are to prevent feature 6 from being too detailed). The results are displayed in Table 3.

Table 3: The score of GPT-4 given specs generated with different configurations.

|  | **RTLLM** | |
| --- | --- | --- |
|  | pass@1 | pass@5 |
| Original NLD | 51.6 | 66.2 |
| SP-Spec-basic | **83.6** | **91.8** |
| SP-Spec-cases | 75.6 | 85.0 |
| SP-Spec-full | 77.2 | 89.0 |

The -case configuration gets the lowest pass rates among GPT-4 specs. We speculate that this is because the examples required by part 5 are presented in multi-modal formats, such as tables or timing diagrams, which are hard for current LLMs to interpret. Meanwhile, the pass rate of the -full configuration is only 2% lower than -basic while still 19% higher than the original NLD. This implies that the accuracy would still be acceptable if the implementation overview is essential (e.g., for large pipelined architectures).

**Answer for Q2**. Examples and implementation overviews may, to some extent, degrade the performance of design agents. However, if the contents are properly restricted, the degree of performance degradation can be controlled.

## 4.4 CASE STUDY 3: ARCHXBENCH LEVEL 4-5

After spec refinement, the accuracy of existing agents over simple tasks is acceptable. So, we explored the limits of our spec *creator* by letting GPT-4 challenge the recently proposed harder benchmark ArchXBench (Purini et al. (2025)). We selected the fixed-point FFT and IFFT cases in level 4 and the Conv2d case in level 5.

When creating specs for FFT and IFFT, we are asked to specify the format of the twiddle factors and decide whether the inputs should be reorganized into bit-reversed order. Then, using this spec as input, 2 out of 5 generation attempts passed the test. This is a plausible progress since the original paper reported a complete failure.

We made similar attempts on the Conv2d case. The original spec requires using shift registers to implement the sliding window to reduce the cost of address decoders. Although the convolution part was implemented correctly, GPT failed to shift the data along the correct dimension or at the correct time. Following our observation that details are harmful, we removed this requirement from the spec. This time, GPT generated an offset-pointer-based implementation. Unfortunately, it failed to build a correct zero-padding mask due to a misunderstanding of the registers' cycle-delay behavior. This is a common issue already observed when generating dividers or floating-point arithmetic units. Nevertheless, although we consider this case a failure, we found the code highly readable and "almost correct", such that a human engineer could fix it in about 10 minutes.

**Answer for Q3**. A better spec can slightly reduce the human workloads on complex designs. However, fine-tuning on tasks that involve multiple submodules or sequential logic still seems essential. We plan as future work to train the models using our refined specs.

## 5 CONCLUSION

A refined spec can help LLMs to produce RTL design more accurately. It should explicitly specify the semantics to avoid mismatches between requirements and test cases. Also, before LLMs are properly trained, users should not guide them to write complex implementations.

LLM has sufficient capabilities (about 90% pass@5 accuracy) to complete simple RTL designs when provided with high-quality specs generated using our *converter* or *creator* tools. Future works that create datasets or benchmarks can use our tools to automatically generate NLDs of higher quality.

## REPRODUCIBILITY STATEMENT

The LLMs used in our experiments (QwenCoder, MG-Verilog, CodeV-R1, GPT-4, Claude 3.5 sonnet) are either open or provide a purchasable API. The prompts of the LLM-based tools (i.e., the *evaluator*, *converter*, and *creator*) are shown in the appendices. The original datasets and benchmarks used in our experiments (i.e., RTLLM, MG-Verilog, VerilogEval, and CodeV-R1) are publicly available, and our refined specs can be reproduced from the provided descriptions.

We plan to release the refined specs under the MIT license.

## ETHICS STATEMENT

This work focuses on improving the quality of natural language descriptions for LLM-based RTL design generation. The experiments use publicly available datasets and do not involve sensitive personal data or harmful model outputs. We do not foresee significant ethical concerns associated with this research.

## LLM USAGE

Large language models (LLMs) are the primary subject of this study. We used LLMs in two ways: (1) as evaluation agents to rate the clarity and simplicity of natural language descriptions (NLDs), and (2) as code-generation agents to test how NLD quality affects design correctness. LLMs were also used to provide suggestions during the exploration of what information an ideal RTL spec should include. However, all decisions regarding methodology design, analysis, and writing were made by the authors.

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

## A PROMPT OF THE EVALUATOR

Below is the prompt we used to evaluate the original NLDs.

```
Please act as a professional Verilog designer. We have designed a
    submodule of our project, and tried to write a spec for it.

Here is the verilog code of the module.

```
{{code}}
```

Here is the spec we wrote.

```
{{spec}}
```

Please help us evaluate, if you want to use our module,
a) Can you learn its exact behavior from the spec without correcting any
    misunderstanding by reading the Verilog code?
b) Can you learn its high-level semantics easily without awareness of its
     internal implementation?

Please score (range from 1 to 5) for each of the questions, and give some
     advice for us to improve. For both questions, a higher score means
    that the spec is easier to read. Please give your response in JSON
    format:

{
    "clarity score": X,
    "simplicity score": Y,
    "suggestions": "..."
}
```

Below is an example of the response of the RTLLM `freq_divbyodd` case.

```
{
    "clarity score": 2,
    "simplicity score": 2,
    "suggestions": "The current spec is too brief and lacks important
        details. It does not describe the timing, duty cycle, phase, or
        waveform characteristics of clk_div. It also omits the behavior
        for different values of NUM_DIV, especially how the output clock
        waveform looks for arbitrary odd divisors. There is no mention of
         the output's duty cycle (e.g., is it 50%?), phase relationship
        to clk, or reset behavior in terms of output state. To improve:
        1) Explicitly describe the output waveform, including duty cycle
        and phase. 2) State how the divider works for any odd NUM_DIV,
        and what happens at reset. 3) Add a timing diagram or example
        waveforms. 4) Specify any constraints or limitations (e.g., only
        odd divisors, minimum/maximum values). 5) Clarify whether NUM_DIV
         is static or can be changed at runtime. This will make the
        module much easier to use as a black box and understand its
        behavior without reading the code."
}
```

## B PROMPT OF THE CONVERTER

Below is the prompt we used to convert RTLLM and VerilogEval descriptions into our specs.

```
702
703  1  Please act as a professional Verilog designer. We have designed a {{name
704     }} module, and we wish to write a spec for it.
705  2
706  3  Here is the verilog code of the module.
     4
707  5  ```
708  6  {{code}}
709  7  ```
     8
710
711  9  Here is the spec we wrote.
     10
712  11 {{spec}}
713  12
714  13 Please help us improve the spec, such that a third-party developer can
715     learn the exact behavior of this module from the spec without reading
716     the verilog code. Please be cautious that the behavior of the code
717     might be unnatural, and the description might be unclear or mistaken.
718     You should strictly follow the code (especially, the output
719     sometimes should be advanced or delayed for several cycles).
     14
720  15 What should be contained:
     16
721  17 - Module name.
722  18 - Behavior details. Please specify the function of this module, i.e., how
723     the output is computed from the input. If it is a sequential logic (
724     e.g., an FSM or pipeline), please specify the timing, i.e., the
725     handshaking protocols, the number of pipeline stages, or the number
726     of cycles before the valid output comes out.
     19 - Parameters. If the module has configurable parameters, please specify
727     their meanings and default values.
728  20 - I/O details. Please specify the semantics of each input or output port.
729     Please detail the meaning of each possible value of the inputs,
730     especially for control signals.
731  21 - Normal cases and corner cases. Please design some cases (input-output
732     pairs) according to the code. We have assured that the code is
733     correct, however its corner case behavior might not have been written
734     in the original spec.
     22 - Implementation overview. Please include some high-level hints of the
735     module's structure. For example, you may describe the semantics of
736     each state in an FSM, each stage in a pipeline, or submodules that
737     might be instantiated.
     23
738  24 What should NOT be contained:
739  25
740  26 - Complicated white-box details. Third-party users wish to treat the
741     module as a black box, and details might be too complex.
742  27 - Trivial corner case behaviors. For example, if the code did not handle
743     overflows, we may just say "assume there are no overflows".
744  28
     29 Please give us the spec in Markdown format.
745
746
747
```

Below is the improved spec for the RTLLM `serial2parallel` case as shown in Figure 2.

```
748  1  # Specification: `serial2parallel` Module
749  2
750  3  ## 1. Module Name
751  4
752  5  **serial2parallel**
753  6
     7  ---
754  8
755  9  ## 2. Functional Overview
     10
```

The `serial2parallel` module converts a stream of serial input bits into an 8-bit parallel output. It collects 8 consecutive serial bits, in order, and then outputs them as an 8-bit parallel value. The module uses a clocked, synchronous design with an active-low reset. The output data is accompanied by a validity signal to indicate when the parallel output is valid.

**Important Note:**
The module's output (`dout_parallel` and `dout_valid`) is only asserted after the **8th valid serial bit is received**. The output is not immediately available after the 8th input; instead, it is presented on the next clock cycle, and the valid signal is asserted for one clock cycle only.

---

## 3. Parameters

- **No configurable parameters.**
  The module is fixed to collect 8 serial bits per parallel output.

---

## 4. I/O Ports and Semantics

| Port Name     | Direction | Width | Description |
|---------------|-----------|-------|-------------|
| clk           | Input     | 1     | Clock signal. All operations are synchronous to the rising edge of `clk`. |
| rst_n         | Input     | 1     | Asynchronous reset, active low. When `rst_n` is 0, all internal state and outputs are reset. |
| din_serial    | Input     | 1     | Serial data input. The value is sampled on the rising edge of `clk` when `din_valid` is high. |
| din_valid     | Input     | 1     | Input data valid signal. When high, indicates that `din_serial` holds a valid bit to be sampled. |
| dout_parallel | Output    | 8     | Parallel output data. Valid only when `dout_valid` is high. |
| dout_valid    | Output    | 1     | Output valid signal. High for one clock cycle when `dout_parallel` contains valid data. |

### Input Signal Details

- **clk**: All state transitions and data sampling occur on the rising edge.
- **rst_n**: When low, the module resets all internal registers and outputs.
- **din_serial**: Carries the serial data bit to be collected.
- **din_valid**:
  - `1`: `din_serial` is sampled and counted toward the 8-bit output.
  - `0`: No data is sampled; the internal counter and buffer are reset.

### Output Signal Details

- **dout_parallel**: Contains the 8 collected serial bits, with the first received bit as the most significant bit (MSB), and the last as the least significant bit (LSB).
- **dout_valid**:
  - `1`: Indicates `dout_parallel` contains valid data. Asserted for **one clock cycle** after 8 valid serial bits are received.
  - `0`: Output is not valid.

---

## 5. Behavioral and Timing Details

### Data Collection and Output Protocol

- The module begins collecting serial bits when `din_valid` is high.
- Each rising edge of `clk` with `din_valid = 1`:
  - The value of `din_serial` is shifted into an internal buffer.
  - An internal counter increments.
- When 8 valid bits have been collected:
  - On the **next clock cycle**, `dout_parallel` outputs the 8-bit value, and `dout_valid` is asserted high for one clock cycle.
  - The output bits are ordered such that the **first serial bit received becomes the MSB** (`dout_parallel[7]`), and the **last bit becomes the LSB** (`dout_parallel[0]`).
- If `din_valid` is low at any time, the internal counter and buffer are reset to zero, and the collection process restarts.

### Reset Behavior

- When `rst_n` is low:
  - All internal state (counter, buffer) and outputs (`dout_parallel`, `dout_valid`) are reset to zero immediately.

---

## 6. Normal and Corner Case Examples

### Example 1: Normal Operation

| Cycle | din_valid | din_serial | Internal Counter | Buffer (MSB..LSB) | dout_valid | dout_parallel |
|-------|-----------|------------|------------------|-------------------|------------|---------------|
| 0     | 0         | X          | 0                | 00000000          | 0          | 00000000      |
| 1     | 1         | 1          | 1                | 00000001          | 0          | 00000000      |
| 2     | 1         | 0          | 2                | 00000010          | 0          | 00000000      |
| 3     | 1         | 1          | 3                | 00000101          | 0          | 00000000      |
| 4     | 1         | 1          | 4                | 00001011          | 0          | 00000000      |
| 5     | 1         | 0          | 5                | 00010110          | 0          | 00000000      |
| 6     | 1         | 1          | 6                | 00101101          | 0          | 00000000      |
| 7     | 1         | 0          | 7                | 01011010          | 0          | 00000000      |
| 8     | 1         | 1          | 8                | 10110101          | 0          | 00000000      |
| 9     | 1         | X          | 1                | X                 | 1          | 10110101      |

- On cycle 9, `dout_valid` is asserted and `dout_parallel` outputs the collected bits: `10110101`.

### Example 2: din_valid Drops Before 8 Bits

| Cycle | din_valid | din_serial | Internal Counter | dout_valid | dout_parallel |
|-------|-----------|------------|------------------|------------|---------------|

```
| 1       | 1          | 1          | 1                | 0          |
    00000000     |
| 2       | 1          | 0          | 2                | 0          |
    00000000     |
| 3       | 0          | X          | 0                | 0          |
    00000000     |
| 4       | 1          | 1          | 1                | 0          |
    00000000     |

- If `din_valid` goes low before 8 bits are collected, the counter and
    buffer reset. The collection process restarts from the next valid bit
    .

### Example 3: Continuous din_valid High

- If `din_valid` remains high, the module will output a valid parallel
    word every 8 cycles, with no gaps, as long as `din_serial` continues
    to provide data.

### Example 4: Reset Behavior

- If `rst_n` is asserted low at any time, all outputs and internal states
    are cleared immediately, regardless of other inputs.

---

## 7. Implementation Briefs

- The module uses:
  - An 8-bit shift register to collect serial bits.
  - A 4-bit counter to track the number of valid bits received.
- Data is shifted in on each clock cycle where `din_valid` is high.
- When the counter reaches 8, the parallel output is presented and the
    valid signal is asserted for one cycle.
- If `din_valid` is ever low, the counter and buffer reset, and the
    collection process restarts.

---

## 8. Summary

- **Collects 8 serial bits (with `din_valid` high) and outputs them as an
    8-bit parallel word.**
- **Output is valid for one clock cycle after the 8th bit is received.**
- **If `din_valid` drops before 8 bits are collected, the process
    restarts.**
- **Reset (`rst_n` low) clears all state and outputs immediately.**

---

**End of Specification**
```

The 9th cycle of Example 1 in the "Normal and Corner Case Examples" section pointed out that every 9th bit will be discarded. 12 out of 20 attempts passed using this spec (pass@5 = 0.99), while 0 out of 20 attempts passed using the original description.

## C  PROMPT OF THE CREATOR

The prompt for the *creator* is similar to the *converter*'s, only appended with the following sentence.

```
Please add placeholders wherever you wish us to add more details.
```

This allows the creator to seek completion for undefined behaviors. Below is the response for the ArchXBench FFT/IFFT task as shown in 4.4, added with placeholders.

```
Here is the improved, professional-grade specification for your module `
    fft16_iterative`, rewritten in clear Markdown format. It captures
    both user-facing semantics and timing behavior without exposing
    excessive implementation detail.

---

# Specification: `fft16_iterative`

## Title

**16-point Iterative Fixed-Point Radix-2 FFT/IFFT Module**

---

## Summary

`fft16_iterative` is a resource-efficient, fixed-size, radix-2 FFT/IFFT
    processor that computes the 16-point Discrete Fourier Transform (DFT)
    or its inverse (IDFT) over complex fixed-point input vectors. It
    uses an **iterative, decimation-in-time (DIT)** architecture to
    minimize area and twiddle storage by reusing a single butterfly unit
    across stages.

This module is designed for use in DSP pipelines and embedded signal-
    processing systems with limited hardware resources, and supports **
    fixed-point input/output**, **IFFT mode**, and **automatic output
    scaling**.

---

## Functional Behavior

### Transform Type

The module supports:

* **Forward FFT** (mode = 0): Compute standard DFT with radix-2 DIT.
* **Inverse FFT (IFFT)** (mode = 1): Use conjugated twiddle factors and
    output normalization by 1/N.

### Timing Protocol

* **Computation begins** on the **rising edge** of `start` (a 1-cycle
    pulse).
* Input data must be **stable at the moment of `start`**.
* Output will become valid after a fixed number of cycles (see below).
* `done` is **asserted high for one clock cycle** to indicate that `
    data_real_out` and `data_imag_out` are valid.
* The module is **sequential**, not pipelined. It processes each FFT in
    isolation.

### Latency

* Latency = `log_2(N) * (N/2)` cycles (for N=16, that is 4 * 8 = **32
    cycles**).
* `done` is asserted **exactly 32 cycles after `start`**.

---

## Module Declaration
```

```verilog
module fft16_iterative #(
    parameter N = 16,
    parameter DATA_W = 12,
    parameter COEFF_W = 16,
    parameter GAIN_W = 4
) (
    input clk,
    input rst,
    input start,
    input mode, // 0: FFT, 1: IFFT
    input signed [DATA_W-1:0] data_real_in [0:N-1],
    input signed [DATA_W-1:0] data_imag_in [0:N-1],
    output signed [DATA_W+GAIN_W-1:0] data_real_out [0:N-1],
    output signed [DATA_W+GAIN_W-1:0] data_imag_out [0:N-1],
    output done
);
```

---

## Parameters

| Name      | Default | Description                                                                 |
| --------- | ------- | --------------------------------------------------------------------------- |
| `N`       | 16      | Transform size (only 16 is currently supported).                            |
| `DATA_W`  | 12      | Bit-width of real/imag input.                                               |
| `COEFF_W` | 16      | Bit-width of twiddle coefficients.                                          |
| `GAIN_W`  | 4       | Maximum bit-growth from internal computation; output width is `DATA_W + GAIN_W`. |

---

## Inputs

| Signal            | Width          | Description                                                            |
| ----------------- | -------------- | --------------------------------------------------------------------- |
| `clk`             | 1              | Clock input.                                                          |
| `rst`             | 1              | Synchronous reset (active high). Aborts current computation and resets FSM. |
| `start`           | 1              | One-cycle pulse to trigger a new computation. Input data must be stable during this pulse. |
| `mode`            | 1              | `0`: FFT; `1`: IFFT.                                                  |
| `data_real_in[N]` | `DATA_W` signed | Real parts of input vector. Indexed from 0 to N-1.                   |
| `data_imag_in[N]` | `DATA_W` signed | Imaginary parts of input vector. Indexed from 0 to N-1.              |

---

## Outputs

```
| Signal               | Width                   | Description

     |

| ----------------- | ---------------------- | ----------------------
     ----------------------------------------------------------- |
| `data_real_out[N]` | `DATA_W + GAIN_W` signed | Real parts of the
    output FFT/IFFT vector.
                                                                            |
| `data_imag_out[N]` | `DATA_W + GAIN_W` signed | Imaginary parts of the
    output FFT/IFFT vector.                                                 |
| `done`            | 1                       | Pulses high **exactly
    32 cycles after `start`**, for one cycle, to signal valid output. |
```

---

## Output Semantics

* **IFFT scaling**: Output is normalized by 1/N (for N=16) using
    arithmetic shift (`>> 4`).
* **Output precision**: The result allows `GAIN_W` bits of headroom to
    avoid overflow due to butterfly additions/multiplications.
* **Output format**: The output order is **bit-reversed** compared to
    natural frequency ordering. You may apply a post-reordering step if
    needed.

---

## Corner Case Behavior

* `start` must not be asserted while a previous FFT is ongoing. If
    asserted early, behavior is undefined.
* If `rst` is asserted during a transform, computation is aborted and
    internal state is reset. No output will be produced for that round.
* If `mode` changes during a transform, the behavior is undefined. `mode`
    must be stable from `start` until `done`.

---

## Example Timing (N = 16)

```
| Cycle | `start` | Action                         |
| ----- | ------- | ------------------------------ |
| 0     | 1       | Input latched, transform starts |
| 1-31  | 0       | Internal butterfly computation |
| 32    | 0       | `done = 1`, output valid       |
```

---

## Implementation Overview

* **Butterfly unit**: Computes radix-2 DIT butterfly:

  * Inputs: two complex numbers + twiddle factor
  * Computes: `X[k] = A + BW`, `X[k+N/2] = A - BW`
* **Control FSM**:

  * Manages stage iteration (0 to log(N)-1)
  * Selects butterfly pairs and ROM twiddles
* **Twiddle ROM**:

  * Precomputed cos/sin values (for FFT)
  * For IFFT, twiddles are complex conjugated
* **Data memory**:

  * Intermediate results stored in dual-ported BRAM or registers
```

```
  * Reused across stages
* **Output scaling**:

  * Optional normalization for IFFT via arithmetic right shift by `log(N)
    `

---

## Test Cases

### Case 1: Impulse Input (FFT)

* Input: `data_real_in[0]=1`, rest zeros; `mode=0`
* Expected Output: `data_real_out[i] = 1`, `data_imag_out[i] = 0` for all
    i

### Case 2: Constant Input (IFFT)

* Input: All `data_real_in[i] = 1`, `data_imag_in[i] = 0`; `mode=1`
* Expected Output: `data_real_out[0]=16`, rest zeros, normalized to `1` (
    after >>4)

### Case 3: Overflow Check

* Input: Mix of +2047 and -2048 at `DATA_W=12`
* Expected Output: No overflow; values within `DATA_W + GAIN_W` signed
    range

---

## To Be Added

* [ ] **Exact internal FSM state list and transitions** (optional for doc
    readers)
* [ ] **Twiddle factor format (e.g., fixed-point Q1.15?)**
* [ ] **Clarify output order (natural vs bit-reversed)**
* [ ] **Specify rounding/truncation method (e.g., truncation vs rounding-
    to-nearest)**
* [ ] **Saturation behavior (if any)**

---

Let me know if you'd like to generate a diagram (FSM or butterfly flow),
    testbench scaffolding, or a verification matrix from this spec.
```

The implementation overview part indicates the function of the butterfly unit. Also, the creator noticed that the output might be bit-reversed, so it added a placeholder and asked us to fill it in.

## D    FAILED CASES ANALYSIS OF GPT-4-SP-SPEC-FULL ON RTLLM

We manually examined the designs of the 50 RTLLM cases generated by GPT-4, and listed our findings in Table 4.

GPT-4 achieves higher accuracy on our spec than on the original NLDs in 20 cases, whereas the original NLDs completely failed in 11 cases. We confirmed that 9 of the completely failed cases and 4 of the less accurate cases are caused by (**D1**). These cases can be fixed or improved with our spec.

Only 1 case failed on our spec but succeeded on the original NLD. We found that the full spec we generated unexpectedly included too many implementation details, and causes (**D2**). When given the basic spec, i.e., removing the optional "implementation overview", GPT-4 passed the test.

Table 4: RTLLM cases in which GPT-4 performed differently given the original NLD and Spec, or both failed. The numbers are the test-passing count among 10 samples.

| Type | Name | Original NLD | Spec-full | FP |
|------|------|:---:|:---:|:---:|
| Fixed | freq_divbyeven | 0 | 10 | ✓ |
| | sequence_detector | 0 | 6 | |
| | signal_generator | 0 | 10 | ✓ |
| | asyn_fifo | 0 | 2 | |
| | radix2_div | 0 | 2 | ✓ |
| | alu | 0 | 10 | ✓ |
| | serial2parallel | 0 | 6 | ✓ |
| | parallel2serial | 0 | 10 | ✓ |
| | pulse_detect | 0 | 4 | ✓ |
| | clkgenerator | 0 | 10 | ✓ |
| | multi_8bit | 0 | 10 | ✓ |
| Improved | barrel_shifter | 2 | 10 | ✓ |
| | LFSR | 2 | 10 | |
| | traffic_light | 6 | 10 | ✓ |
| | freq_divbyfrac | 2 | 8 | |
| | fixed_point_substractor | 4 | 8 | |
| | JC_counter | 4 | 10 | ✓ |
| | multi_booth_8bit | 2 | 10 | |
| | freq_div | 4 | 10 | ✓ |
| | multi_pipe_4bit | 8 | 10 | |
| Both failed | float_multi | 0 | 0 | |
| | freq_divbyodd | 0 | 0 | |
| | pe | 0 | 0 | |
| Weakened | div_16bit | 2 | 0 | |

In the other 3 cases, GPT-4 failed on both specs and original NLDs. This indicates that, although a better spec indeed yields a large improvement, enhancing LLMs' understanding of sequential behavior remains necessary.

