# OpenReview forum: "Refining Specs For LLM-Based RTL Agile Design"
_ICLR.cc/2026/Conference — Submitted to ICLR 2026_

### Official Review · Reviewer_mB8U · 2025-10-28

**Soundness:** 2
**Presentation:** 2
**Contribution:** 2
**Rating:** 4
**Confidence:** 3

**Summary:**

This paper addresses the low functional accuracy of large language models (LLMs) in register-transfer-level (RTL) hardware design. The author presents an insightful perspective: the primary reason for the low accuracy is not the model's limitations, but rather the poor quality of the natural language descriptions (NLDs) used in existing benchmark tests. This issue is particularly evident in the mismatch between NLDs and the ground truth behavior, leading to false negatives, as well as overly detailed descriptions. To tackle this problem, the author proposes the "Refined Spec" paradigm, emphasizing the importance of providing clear, well-structured specification documents that focus on behavioral descriptions. Experimental results indicate that the quality of input specifications is crucial for the performance of LLMs in RTL design applications.

**Strengths:**

Novel insights. The paper shifts the research focus from the costly fine-tuning of large language models (LLMs) to the quality of input specifications (prompts), which offer greater engineering value. This finding is fundamentally important for advancing the practical application of LLMs in hardware design.

Effective Guidance. The paper not only identified the problem but also provided a clear structure for a "refined specification" comprising six key components for future dataset creators and RTL engineers, which has significant practical value.

**Weaknesses:**

The details of the automated generation tool are lacking. A core contribution of the paper is the proposed LLM-driven specification refinement toolchain. However, the specific prompts used for these tools, the detailed internal reasoning steps, and the descriptions of the automated workflows are too brief. This absence of essential information significantly undermines the reproducibility of the research findings.

The quantitative support for complex designs is lacking. While the author references the potential of complex tasks such as FFT and Conv2d, there is a notable absence of detailed quantitative pass@k rates for these large-scale, multi-module tasks. The available data are confined to simple modules, hindering a clear demonstration of this method's generalization capability in addressing complex real-world problems.

**Questions:**

Scale capacity. Please provide detailed quantitative pass@k rates for complex tasks such as FFT or Conv2d to demonstrate the effectiveness of this method and its ability to be extended to large-scale designs.

Data leakage. Please discuss how to ensure that, during the generation of the refining specification, the GPT-4 used does not indirectly utilize the ground truth values of the original benchmark that may be present in the pre-training data.

Generalization. Can the specification's validity generalize across models, including open-source and closed-source models such as the Llama series and Claude?

---

> ### Author Response · Authors · 2025-11-15
> **Author response for reviewer mB8U**
>
> Dear reviewer mB8U,
>
> Thank you for your comprehensive comments and suggestions. While extended experiments are underway, we hereby provide some clarifications that may be of interest to you.
>
> **For weakness 1**: Due to the page limit, we only kept the key findings in the main text (Section 3). The prompts we used in our experiments, and examples of LLM responses, are shown in the appendix. Our experiment results can be reproduced with these prompts.
>
> **For questions 2 and 3**: For the spec generation task, the original NLDs and the GM codes of the benchmarks are intentionally provided as part of the input. However, what we asked the LLMs to perform is a totally different task, i.e., to generate the "refined specs" which never exist. So, the generation process fully relies on the generalization ability of the model, and there is no data leakage.
>
>  For the code generation task, we observed that, while GPT-4-SP has similar results as previously reported, GPT-4-SP-Spec-full (i.e., using our spec as input) outperforms other configurations. If the training data of GPT-4 contains the benchmark, GPT-SP should have similar accuracy as GPT-4-SP-Spec-full.
>
> We will present more experiment results on cross generation: use one model to create the spec, and another model to generate code. We expect this can show the generalizability, and also prevent the model from "embedding" any information, which only itself can understand, into the spec.

---

> > ### Comment · Reviewer_mB8U · 2025-11-28
> >
> > The author did not address my concerns, such as quantitative pass@k rates for complex tasks like FFT or Conv2d, and the validity generalization across models. Therefore, I will maintain my score.

---

> ### Author Response · Authors · 2025-11-28
>
> Dear reviewer mB8U,
>
> We have conducted the cross-evaluation experiments to show the performance of specs (generated by GPT-4 and Claude 3.5) on code generation models (QwenCoder, CodeVR1, Claude 3.5, and GPT 4). We suggest that the results (shown in Table 2 of the revised paper) may address your concern on generalization.
>
> In the meantime, we are preparing more experiments for complex tasks (ArchXBench). According to our *creator* tool, the original specs of ArchXBench requires manual completion, because the background knowledge introductions are insufficient, and the benchmark did not provide GMs/FVs for the LLMs to infer the behavior. This took us more time, so we temporarily kept the original version of Section 4.4. Updates will be made as soon as possible.

---

### Official Review · Reviewer_3WLA · 2025-10-29

**Soundness:** 3
**Presentation:** 2
**Contribution:** 2
**Rating:** 4
**Confidence:** 4

**Summary:**

This paper investigates the low functional correctness of LLM-generated RTL designs, positing that this inaccuracy primarily stems from the poor quality and ambiguity of input specifications (specs). The authors identify two critical shortcomings in existing specifications: first, they are often too vague, leading to functional mismatches with golden models (GMs) and numerous false negatives during testing; second, some are overly detailed, effectively serving as direct translations of the implementation, which fail to properly evaluate the LLM’s high-level design capability. To address these issues, this work proposes a new framework for refining specifications by explicitly incorporating essential information such as I/O semantics, formulas, and sequential features, thereby minimizing behavioral discrepancies between the spec and the GM. Empirical results confirm the efficacy of this approach: when provided with these high-quality specs, general-purpose LLMs achieve substantially improved performance, with GPT-4 attaining a pass@5 score of 89.0% on RTLLM and 96.0% on VerilogEval-Human, suggesting that current LLMs are sufficiently capable of generating small RTL modules given precise requirements.

**Strengths:**

The problem addressed in this work is of considerable importance, as enhancing the accuracy of LLM-generated RTL designs could substantially shorten the hardware development cycle. This study’s primary contribution lies in its insightful shift in perspective, revealing that the low functional accuracy of LLM-generated RTL designs stems primarily from the poor quality of input specifications (specs). The identified phenomenon, i.e., the mismatch between specs and golden models (GMs), provides a crucial and actionable insight for constructing more reliable benchmarks and datasets in the future. Building on this diagnosis, the authors propose a simple yet effective solution: a refined specification framework that explicitly incorporates essential sequential and semantic features. The experimental results are compelling, demonstrating that with these refined specifications, general-purpose LLMs, e.g., GPT-4, achieve significantly improved performance in RTL design generation.

**Weaknesses:**

While the main idea of this paper is insightful, several weaknesses diminish its overall academic rigor and impact. Foremost, the manuscript contains numerous typographical and grammatical errors (*e.g.*, the undefined acronym “LOC” in the Abstract and grammatical issues such as “their capability have...” on Line 34), which indicate a lack of thorough internal review and undermine the paper’s professional presentation. In addition, for ICLR submissions, the Abstract must be limited to a single paragraph, but the current version spans multiple paragraphs. Furthermore, the overall formatting and adherence to conference standards are suboptimal. The authors frequently misuse citation commands (*e.g.*, employing `\citet` where `\citep` is required by ICLR style) and omit mandatory or strongly recommended statements regarding ethics, reproducibility, and LLM usage.

From a technical standpoint, the novelty appears limited. The proposed specification format, which primarily extends existing structures by incorporating sequential and semantic features, represents an incremental improvement, particularly since the authors themselves acknowledge its structural similarity to the manually constructed RealBench format. While employing an LLM-based converter to refine specifications adds convenience, the refinement methodology itself appears technically straightforward and may not meet the innovation threshold expected at ICLR.

Moreover, the experimental validation lacks robustness and completeness. Evaluating the method on only two general-purpose LLMs restricts the generalizability of the findings, and the absence of discussion regarding the unexpected decline in Qwen-2.5-Coder’s pass@1 score (Table 1) introduces ambiguity that weakens the credibility of the central claim. To substantiate their conclusions, the authors should include additional experiments across a broader range of models, including those fine-tuned specifically for RTL generation.

Finally, the Related Work section is insufficiently detailed. Although key dataset papers such as MG-Verilog, CodeV, and CraftRTL are cited, the authors fail to provide a substantive comparative discussion of these works’ data construction methodologies relative to their own LLM-based specification refinement. This omission represents a missed opportunity to situate the contribution within the broader landscape and to clarify its distinct advantages and limitations.

**Questions:**

1. The paper presents a contradictory tension regarding the appropriate level of detail in specifications. On one hand, it argues that specifications should be "not detailed in implementation" and over-detailed NLDs are detrimental to benchmarking (Section 3.2). On the other hand, the proposed refinement requires the inclusion of highly specific information like "sequential features", "formulas", and precise "I/O signal semantics" (Section 3.3). What is the precise boundary or heuristic distinguishing beneficial, function-defining semantic detail (which the LLM requires) from detrimental, implementation-dictating detail (which confuses the LLM)? Clarifying this "balancing point" is crucial for future users of the specification framework.
2. Table 2 shows that adding theoretically helpful components, such as examples (`Spec-cases`) and implementation briefs (`Spec-full`), actually decreases performance relative to the minimal `Spec-basic` configuration. This raises the question of what constitutes the best practical specification format. A more thorough ablation study is needed to determine which sub-components (e.g., Declaration, Behavior details, Sequential features, I/O details) are individually most beneficial.
3. The current application of the "converter" tool relies on the existence of a golden model (GM) to identify missing requirements and correct misalignments between the specification and the GM/Testbench. If a designer or dataset curator wishes to apply the specification refinement methodology to a large-scale training dataset where GMs are *not* available, how should the refinement process be executed? Will models fine-tuned purely on a refined training dataset (without a debugging loop) demonstrate the same magnitude of performance improvement observed during benchmarking?
4. The paper highlights the difficulty fine-tuned models have with high-level tasks despite training on low-level examples (Section 3.2). Notably, DeepRTL[1] employs a curriculum learning strategy, training on low-level prompts before proceeding to high-level ones. Could a similar curriculum approach mitigate the challenge identified in this work? Additionally, given the focus on better alignment between code and natural language, how does the structure and content of the final refined specification compare to the natural language descriptions constructed by DeepRTL, MG-Verilog, or CraftRTL? A detailed comparison is needed to establish novelty in this crowded research area.

    [1] Liu, Y., Changran, X. U., Zhou, Y., Li, Z., & Xu, Q. DeepRTL: Bridging Verilog Understanding and Generation with a Unified Representation Model. In *The Thirteenth International Conference on Learning Representations*.

5. The main results demonstrate performance gains for general-purpose LLMs (GPT-4 and Qwen2.5-Coder) when using the refined specification. To confirm the robustness and broader applicability of the refined specification, have the authors evaluated their specifications with LLMs specifically fine-tuned for RTL generation (*e.g.*, the CodeV-R1 or CraftRTL models mentioned in Table 1)?

---

> ### Author Response · Authors · 2025-11-14
> **Author response for reviewer 3WLA (Part 1)**
>
> Dear reviewer 3WLA,
>
> Thank you for your comprehensive comments and suggestions. While extended experiments are underway, we hereby provide some clarifications that may be of interest to you.
>
> **For weakness 1**: Thank you for your instruction. These errors will be fixed in the upcoming revision.
>
> **For weakness 3 and question 5**: We will present more experiment results on cross generation: use one model to create the spec, and another model to generate code. We expect this can show the generalizability, and also prevent the model from "embedding" any information, which only itself can understand, into the spec.
>
> **For question 1**: Very good question. Here, we clarify the difference between the two types of details. Semantic details describe the interfaces of the module as a **black box**. If the user replaces the module with another implementation with the same semantics, the system should work without any further changes. This is why the semantics are essential. We have stated in the paper that semantic details consist of (at least):
> - What is the signature of the module (i.e., module name and port declaration)?
> - What is the relationship between the output and the input?
> - How many cycles should the user wait before receiving the output, or which signal identifies a valid output?
> - Is the module pipelined, or can it accept a set of inputs while processing another one?
> - What are the meanings of each I/O port?
>
> According to this principle, the definition of *a part of* the states of an FSM, although intuitively more like implementation details, should be regarded as semantics. For example, for a serial-to-parallel module, it is essential to specify whether a frame contains a starting bit or parity bits. It is convenient to describe the meaning of each bit as an FSM state.
>
> On the other hand, implementation details describe the internal micro-architecture as a **white box**. The micro-architecture of a module can be optimized for better PPA, as long as the semantics are kept unchanged.
>
> Existing LLMs are trained mainly on software codes and thus are unfamiliar with micro-architecture-level instructions. Without the implementation details, the LLMs may present functionally correct HDL codes "in software style", e.g., using address decoders to access arrays instead of shift registers, although the latter can be much cheaper. With *very few* implementation details, the LLMs generate erroneous designs. With *complete* implementation details, the code generation task degenerates into a line-by-line translation task. This is why most models perform better on the VerilogEval-Machine benchmark. Nevertheless, as we have stated in the paper, it is impractical for users to provide complete details. So at least *for now*, before the LLMs are properly trained, the implementation details are detrimental. We suggest that users should exclude these from their spec:
> - What internal registers should be involved?
> - How should the function be partitioned into pipeline stages?
>
> However, users can manually write the definition of registers and ask the LLMs to fill in the combinational logic between them. Combinational logic is more like pure functions in software code, in which wires are similar to immutable variables, so the LLMs are more familiar with this style.
>
> We will update Section 3.3 (SPEC GENERATION) to present our findings more clearly.

---

> ### Author Response · Authors · 2025-11-14
> **Author response for reviewer 3WLA (Part 2)**
>
> **For weakness 2**: We admit that our methodology of refining specs is straightforward. However, our motivation is to point out that a clearer spec is very helpful for the LLM RTL design agents, and this clearer spec can actually be presented at low cost (i.e., compared with self-fixing). Although works on finetuning and self-fixing are also very beneficial, the accuracy can be further improved, or the computing power consumption can be lowered with a better input.
>
> We indeed acknowledge the similarity in format between our specs and those of RealBench. But still, we have good reasons to present our results to the community.
> - RealBench is a new benchmark, and only contains specs for new cases. Our refined specs for old cases are more comparable with existing works.
>     - Our results confirmed the capability of existing LLMs on simple tasks.
> - We discussed what a good spec should contain. While RealBench aims to explore the abilities of the LLMs, our results point the way for future benchmark (and training dataset) creators.
>     - In fact, RealBench contains too many implementation instructions, which we have discovered to be detrimental. It also contains non-textual components (e.g., block diagrams and data flow diagrams) that we are uncertain whether the LLMs can interpret. So, it is possible that the RealBench specs can be further refined with our framework. We have to leave this as future work due to the limited time.
> - Also, we suggest that the specs created by the LLMs are probably most suitable for themselves to interpret.
>
> **For question 2**: As pointed out in our response to question 1, **all** parts of our Spec-basic (that is, the semantics) are essential. Obviously, at least all the constraints involved in the TB, or the formal verification assertion set (FV), must be mentioned, otherwise the designer (either human or LLM) has to guess the desired behavior.
>
> However, as we have discussed in Section 3.5, users can remove unnecessary constraints to simplify the task. For example, LLMs seem to prefer valid-ready handshaking protocols, such that they are free to choose the number of cycles for each operation, rather than wait for a fixed number of cycles. So, if the performance is not sensitive, users can replace the sequential feature constraints with handshaking signal definitions.
>
> **For question 3**: To perform supervised training, each case in the dataset must have either a GM, **a TB, or an FV**. If the case has a GM, one can use fuzzing techniques and diff-test for verification. If the case has a TB or an FV, one can directly use the TB or the FV for verification. If all of these are missing, one cannot verify whether the generated design is correct, and this case cannot be used for training.
>
> The *converter* tool is designed to remove the **mismatch** between the NLDs and the GMs / TBs / FVs. In Section 2, we have demonstrated the RTL design workflow. The designer first present the NLD, then formalize it as a TB or FV, and finally present a semantically equivalent GM. During this process, the designer adds more information step-by-step into the design, which causes the mismatch between the final outcome and the original NLD. Theoretically, any of these three can be used as input. However, in the benchmarks we are using, the TBs are too simple for the LLMs to infer the expected behavior.

---

> ### Author Response · Authors · 2025-11-16
> **Author response for reviewer 3WLA (Part 3)**
>
> **For question 4, part 1**: We suggest that the potentials of the curriculum learning strategy are not fully explored. MG-Verilog and DeepRTL are the representative works in this category. MG-Verilog is evaluated under the VerilogEval benchmark. The paper reported a 55% (given only high-level summaries) - 60% (given detailed summaries) correctness rate, which is lower than other finetuned small models (as listed in our Table 1). Meanwhile, DeepRTL is evaluated under the benchmark presented in the paper "Natural language is not enough" [1], which is another extension of RTLLM. According to the paper, DeepRTL-16B still fails on simple Arithmetic and Logic cases after finetuning, while other finetuned models can pass these cases easily. Until the submission of our paper, the Github repo of DeepRTL is still empty, so we cannot reproduce their experiments.
>
> [1] Chang, Kaiyan, et al. "Natural language is not enough: Benchmarking multi-modal generative AI for Verilog generation." Proceedings of the 43rd IEEE/ACM International Conference on Computer-Aided Design. 2024.
>
> **For question 4, part 2**: In Sections 3.1 and 3.2, we pointed out two major issues of existing datasets: NLD-GM mismatch and complicated implementation details. We observed that these issues remain in recent LLM-created datasets, including those used to train MG-Verilog, CraftRTL, and CodeV-R1.
>
> The major cause of the first issue is that when creating the descriptions (high-level global summaries), the LLMs are usually asked to "summarize the function" of the module, resulting in brief summaries. Here is an example (id: 105513) from CodeV-R1 dataset.
>
> ```
> Design a Verilog module that generates the horizontal and vertical synchronization signals for a VGA display. The module should also output the current horizontal and vertical counters. The module should use parameters to define the VGA timing parameters, including width, height, front porch, pulse width, and back porch for both horizontal and vertical synchronization. The module should operate on a clock signal and a clear signal, and it should increment the horizontal and vertical counters on each clock cycle. The counters should reset when they reach their maximum values or when the clear signal is active. The synchronization signals should be generated based on the current values of the counters, ensuring that the VGA display is properly synchronized. This Verilog module, named VGA_timings, has the interface designed as follows: ...
> ```
>
> We can see that the description said the horizontal and vertical counters should be increased per cycle, but if the reader is not familiar with VGA, they cannot figure out how the counters and the sync signals work together. This issue also occurs in the high-level and medium-level block descriptions of DeepRTL (the only example is which displayed in the paper).
>
> The major cause of the second issue is, as we mentioned in the paper, when the LLMs are asked to provide "details", they tend to give a line-by-line translation. Real-world specs and comments will not be written like this. Here is an example (id: 2456) of the block summaries from the MG-Verilog dataset.
>
> ```
> wire signed [14:0] x2 = pre<<<1;
> wire signed [15:0] x3 = x2+pre;
> wire signed [15:0] x4 = pre<<<2;
> wire signed [16:0] x6 = x4+x2;
> wire signed [16:0] x8 = pre<<<3;
> wire signed [17:0] x12 = x8+x4;
> wire signed [17:0] x16 = pre<<<4;
> always @(posedge clk)
> if( rst )
> post <= 16'd0;
> else
> if( sample )
> case( volume )
> 3'd0:
> post <= { {2{pre[13]}}, pre };
> 3'd1:
> post <= { x2[14], x2 };
> 3'd2:
> post <= { x2, 1'd0 };
> 3'd3:
> post <= x4;
> 3'd4:
> casex( x6[16:15] )
> 2'b00, 2'b11: post <= x6[15:0];
> 2'b0x: post <= 16'h7FFF;
> 2'b1x: post <= 16'h8000;
> endcase
> ...
>
> block_0: The block_0 code block performs left shift, right shift, and addition operations on the pre input signal, resulting in the x2, x3, x4, and x6 signals.
> block_1: block_1 performs left shift, addition, and right shift operations on the pre input signal, resulting in the x8, x12, and x16 signals. It also contains an always block triggered by the positive edge of the clk signal, with a reset condition if rst is true.
> block_2: The block_2 code block assigns a reset value to the post signal, and then checks for the sample condition. If the sample condition is true, it selects a volume based on the input volume signal.
> block_3: The block_3 code block assigns the output post signal based on the selected volume. If volume is 0, it assigns the post signal with the leftmost 2 bits of the pre signal followed by the pre signal itself. If volume is 1, it assigns the post signal with the leftmost bit of the x2 signal followed by the x2 signal itself. If volume is 2, it assigns the post signal with the x2 signal followed by a 0 bit.
> ...
> ```
>
> Although it claims to be "summary", the block-level descriptions seem to be translating the code. The reasoning gap between the high-level and low-level descriptions is hardly solved.

---

> ### Author Response · Authors · 2025-11-16
> **Author response for reviewer 3WLA (Part 4)**
>
> To provide more statistical insights, we used our *evaluator* tool to measure the quality of original RTLLM descriptions, the MG-Verilog descriptions, and our refined RTLLM specs. Each case is given two scores, the clarity score and the black box score, that measures:
>
> - Can you learn its exact behavior from the spec without reading the verilog code?
> - If you wish to use this model as a black box, can you learn the high-level semantics easily without studying the details?
>
> The score distribution of original RTLLM NLDs is:
>
> | clarity \ black box | 1 | 2 | 3 | 4 | 5 |
> |---|---|---|---|---|---|
> | 1 | 0.0 | 0.0 | 0.0 | 0.0 | 0.0 |
> | 2 | 0.0 | 20.0 | 22.0 | 0.0 | 0.0 |
> | 3 | 0.0 | 4.0 | 16.0 | 20.0 | 0.0 |
> | 4 | 0.0 | 0.0 | 0.0 | 14.0 | 4.0 |
> | 5 | 0.0 | 0.0 | 0.0 | 0.0 | 0.0 |
>
> The score distribution of MG-Verilog NLDs is:
>
> | clarity \ blackbox | 1 | 2 | 3 | 4 | 5 |
> |---|---|---|---|---|---|
> | 1 | 0.0 | 0.5 | 0.0 | 0.0 | 0.0 |
> | 2 | 2.0 | 20.3 | 33.0 | 0.0 | 0.0 |
> | 3 | 0.0 | 5.6 | 1.5 | 22.8 | 0.0 |
> | 4 | 0.0 | 0.0 | 0.5 | 7.6 | 4.1 |
> | 5 | 0.0 | 0.0 | 0.0 | 0.0 | 2.0 |
>
> The score distribution of CodeV-R1 NLDs is:
>
> | clarity \ black box | 1 | 2 | 3 | 4 | 5 |
> |---|---|---|---|---|---|
> | 1 | 0.0 | 0.0 | 0.0 | 0.0 | 0.0 |
> | 2 | 0.0 | 4.0 | 9.0 | 1.0 | 0.0 |
> | 3 | 0.0 | 9.0 | 9.0 | 25.0 | 0.0 |
> | 4 | 0.0 | 0.0 | 2.0 | 29.0 | 11.0 |
> | 5 | 0.0 | 0.0 | 0.0 | 0.0 | 1.0 |
>
> The score distribution of the RTLLM refined specs is:
>
> | clarity \ black box | 1 | 2 | 3 | 4 | 5 |
> |---|---|---|---|---|---|
> | 1 | 0.0 | 0.0 | 0.0 | 0.0 | 0.0 |
> | 2 | 0.0 | 0.0 | 0.0 | 0.0 | 0.0 |
> | 3 | 0.0 | 2.0 | 0.0 | 0.0 | 0.0 |
> | 4 | 0.0 | 0.0 | 0.0 | 8.0 | 26.0 |
> | 5 | 0.0 | 2.0 | 0.0 | 2.0 | 60.0 |
>
> The number in the tables are percentages. It is obvious that the evaluator (based on GPT-4) thinks that the refined Specs are more suitable for it to understand. Because the CodeV-R1 dataset does not contain any low-level descriptions, it has high black box score, but its clarity score lies mainly between 3 and 4.
>
> These tables will be added to the paper as figures.

---

### Official Review · Reviewer_9Vxd · 2025-10-30

**Soundness:** 2
**Presentation:** 2
**Contribution:** 2
**Rating:** 2
**Confidence:** 3

**Summary:**

This paper investigates the impact of natural language specification (NLD) quality on the performance of large language models (LLMs) for register-transfer-level (RTL) hardware code generation tasks. The authors identify sources of functional failures in existing benchmarks, attributing many to ambiguous or over-detailed prompts rather than core LLM architectural limits. They propose tools and a guided workflow to generate well-structured, human-friendly specs, demonstrate how improved specs can drastically boost LLM code correctness (e.g., GPT-4 pass@5 up to 96% on VerilogEval-Human), and analyze the interplay between spec configurations and empirical results.

**Strengths:**

- Through targeted case studies, the paper documents how functional errors in generated RTL code are often due to missing or misaligned requirement details in the NLDs, not the inability of LLMs to handle the tasks.
- The experiments show that proper specs can raise GPT-4 performance from ~55% to over 96% pass@5 on key benchmarks. This is an impressive improvement achieved with minimal additional annotation cost.
- The tools—evaluator, converter, and creator—are practical. Detailed guides and templates are provided, demonstrating reproducibility and immediate usability for both dataset creators and practitioners.

**Weaknesses:**

- The main contribution is in process/tooling/template and benchmarking methodology; there is no new LLM architecture, optimization method, or theoretical framework. While impactful in improving practical results, the intellectual advancement is more incremental and process-focused rather than fundamental.
- Although the converter/creator tools are claimed to automate spec generation, the actual failure cases, limits of LLM-assistance for complex sequential logic, and required human intervention are only anecdotally illustrated. There is arguably an overestimate of how easily these tools can scale
- The experiments focus mainly on GPT-4 (with some Qwen2.5 comparisons). Robustness to other LLMs, out-of-domain specs, dataset noise, or alternative code synthesis settings is underexplored.
- There is no explicit discussion of code/data/method availability. While the methodology could be reconstructed from the paper, an explicit open-sourcing statement or link would help adoption.
- While the work attributes errors to false negatives due to behavioral/mismatch in specs vs. testbenches, quantitative breakdowns (percentages, confusion matrices, etc.) across all classes of failures are sparse. The manual investigation in Section 3.1 notes "over 30%" FNs but lacks further statistical rigor.

**Questions:**

- Please provide more detailed error/diagnostic analysis for false negatives/positives across RTLLM and VerilogEval—could you present confusion matrices or breakdowns by error type to clarify which failures are fixable by spec alone?
- Are there plans to release your evaluator/converter/creator tools and enriched benchmarks, and if so, under what license?
- Can the authors offer empirical insights (e.g., via ablation/connectivity experiments) into how well the methodology generalizes to other LLMs (e.g., Gemini, LLama)? Does the gain from spec refinement persist across backbone LLMs?
- How practically usable is the creator tool for large/enterprise-scale RTL projects, especially for spec aspects that cannot be easily inferred without significant human domain expertise?
- Beyond functional correctness, what is the observed impact of improved specs on physical design metrics (e.g., timing, power, area) if any? Do more precise specs bias LLMs toward certain microarchitectures or synthesis results?

---

> ### Author Response · Authors · 2025-11-13
> **Author response for reviewer 9Vxd (Part 1)**
>
> Dear reviewer 9Vxd,
>
> Thank you for your comprehensive comments and suggestions. While extended experiments are underway, we hereby provide some clarifications that may be of interest to you.
>
> **For weakness 1**: We admit that our methodology of refining specs is straightforward. However, our motivation is to point out that a clearer spec is very helpful for the LLM RTL design agents, and this clearer spec can actually be presented at low cost (i.e., compared with self-fixing). Although works on finetuning and self-fixing are also very beneficial, the accuracy can be further improved, or the computing power consumption can be lowered with a better input.
>
> **For weakness 4 and question 2**: The prompts we used in our experiments, and examples of LLM responses, are shown in the appendix. Our experiment results can be reproduced with these prompts. Still, we plan to make our results open-source under MIT license.
>
> **For weakness 2, 5 and question 1**: In this topic, *false negative* refers to the situation that the LLMs are misled by low quality NLDs, and *false positive* refers to the situation that the TBs are too simple to discover errors in LLM created designs. Our work focus on the FNs, while FPs are studied by existing works, such as [1].
>
> It is hard to formally quantify whether an informal NLD matches the behavior of a piece of formal Verilog code. However, we can show some insights based on human inspection. Among the 50 cases of RTLLM, the following 11 cases are which GPT-4 failed when given the original spec, but succeed when given our refined spec:
>
> - freq_divbyeven: (1.sp)=0, (2.sp+spec)=10
> - sequence_detector: (1.sp)=0, (2.sp+spec)=6
> - signal_generator: (1.sp)=0, (2.sp+spec)=10
> - asyn_fifo: (1.sp)=0, (2.sp+spec)=2
> - radix2_div: (1.sp)=0, (2.sp+spec)=2
> - alu: (1.sp)=0, (2.sp+spec)=10
> - serial2parallel: (1.sp)=0, (2.sp+spec)=6
> - parallel2serial: (1.sp)=0, (2.sp+spec)=10
> - pulse_detect: (1.sp)=0, (2.sp+spec)=4
> - clkgenerator: (1.sp)=0, (2.sp+spec)=10
> - multi_8bit: (1.sp)=0, (2.sp+spec)=10
>
> (1.sp) means GPT-4 with self-plan enabled, given the original description. (2.sp+spec) means given our spec. The number is the pass count out of 10 samples.
>
> 9 of the cases, namely {freq_divbyeven, signal_generator, alu, serial2parallel, parallel2serial, pulse_detect, clkgenerator, multi_8bit, radix2_div}, are confirmed to have false negatives: missing definition of params, ports or initial values caused the failure. {sequence_detector, async_fifo} are complex sequential logic. Our spec did not provide guidance on how to implement these modules, but still, a clearer spec can slightly increase the probability of generating correct designs.
>
> The following are which both configuration failed:
>
> - float_multi: (1.sp)=(2.sp+spec)=0
> - freq_divbyodd: (1.sp)=(2.sp+spec)=0
> - pe: (1.sp)=(2.sp+spec)=0
>
> These 3 cases show that the improvements in accuracy is indeed limited.
>
> Only one case failed when using our spec, while succeeded using the original one:
>
> - div_16bit: (1.sp)=2, (2.sp+spec)=0
>
> We found that the spec we generated unexpectedly included too many implementation details. As we have concluded in the paper, these details are actually detrimental for code generation tasks. When given the Spec-basic version, i.e., without the implementation section, GPT-4 passed the test.
>
> There are more cases that GPT-4 succeeded given both version of NLDs, but the success rate given the original NLD is much lower:
>
> - barrel_shifter: (1.sp)=2 < (2.sp+spec)=10
> - LFSR: (1.sp)=2 < (2.sp+spec)=10
> - traffic_light: (1.sp)=6 < (2.sp+spec)=10
> - freq_divbyfrac: (1.sp)=2 < (2.sp+spec)=8
> - fixed_point_substractor: (1.sp)=4 < (2.sp+spec)=8
> - JC_counter: (1.sp)=4 < (2.sp+spec)=10
> - multi_booth_8bit: (1.sp)=2 < (2.sp+spec)=10
> - freq_div: (1.sp)=4 < (2.sp+spec)=10
> - multi_pipe_4bit: (1.sp)=8 < (2.sp+spec)=10
>
> Among these, 4 more cases, namely {barrel_shifter, traffic_light, JC_counter, freq_div} are also confirmed to have false negatives. However, it is possible for the LLMs to guess the correct behavior, so the pass count is not 0. For example, the NLD of the barrel_shifter case did not mention whether to shift left or right. GPT-4 has 1/2 probability to shift right, which is the expected behavior.
>
> We will put these results in the appendix due to the page limit. Also, we have some statistical results produced by our *evaluator* tools, showing the distribution of clarity and simplicity scores of the original NLDs and our specs. These results can be found in our **Author response for reviewer 3WLA (Part 4)**, and will be added to the paper as figures.
>
> [1] Jin, Pengwei, et al. "Realbench: Benchmarking verilog generation models with real-world ip designs." arXiv preprint arXiv:2507.16200 (2025).

---

> ### Author Response · Authors · 2025-11-13
> **Author response for reviewer 9Vxd (Part 2)**
>
> **For weakness 3 and question 3**: We will present more experiment results on cross generation: use one model to create the spec, and another model to generate code. We expect this can show the generalizability, and also prevent the model from "embedding" any information, which only itself can understand, into the spec.
>
> **For question 4**: The creator does not "generate" descriptions; it guides the users to describe their needs clearly. In other words, human expertise is essential. However, the creator can still reduce human efforts, because users only have to describe the functionality instead of connecting hundreds of wires.
>
> For enterprise-scale RTL projects, AutoSilicon [1] proposed a self-planning strategy that automatically performs this decomposition. Theoretically, combining [1] and our work, the description of the submodules can further be refined with our creator tool. We leave this as future work.
>
> [1] Li, Cangyuan, et al. "Autosilicon: Scaling up RTL design generation capability of large language models." ACM Transactions on Design Automation of Electronic Systems 30.6 (2025): 1-21.
>
> **For question 5**: Adding implementation details for potentially better PPA currently harms the correctness rate, so as we have discussed in Section 3.5, "it is too early to optimize PPA". We think PPA optimizations should be conducted at least after all the cases in RTLLM and VerilogEval can be correctly completed by LLMs, since they are indeed simple (for human designers).
>
> Despite this, the LLM-generated designs with unsatisfactory PPA can still be used for agile deployment of FPGA-based services, as long as they are functionally correct. Accelerators, such as the systolic arrays and spatial accelerators mentioned by [2], leverage hardware for parallelization. As long as the design can be deployed onto the FPGA without timing or power violations, the PPA is not the user's main concern. Meanwhile, trusted computing services rely on hardware-based security features (e.g., physical isolation [3]) and primitives (e.g., oblivious memory access [4]). The quicker the security features are deployed, the lower the risk for the threatened sensitive applications. This was our initial motivation for starting our LLM agile design research.
>
> [2] Liu, Yi, et al. "Deeprtl: Bridging Verilog understanding and generation with a unified representation model." arXiv preprint arXiv:2502.15832 (2025).
>
> [3] Zhao, Mark, Mingyu Gao, and Christos Kozyrakis. "Shef: Shielded enclaves for cloud FPGAs." Proceedings of the 27th ACM International Conference on Architectural Support for Programming Languages and Operating Systems. 2022.
>
> [4] Wang, Xiao, Hubert Chan, and Elaine Shi. "Circuit oram: On tightness of the Goldreich-Ostrovsky lower bound." Proceedings of the 22nd ACM SIGSAC Conference on Computer and Communications Security. 2015.

---

> > ### Comment · Reviewer_9Vxd · 2025-11-26
> >
> > Thank you for the detailed response! I increased my score accordingly.
> >
> > I appreciate the specific breakdown on error analysis of the RTLLM cases provided in the rebuttal and Appendix D. This addresses my previous concern regarding the lack of diagnostic rigor.
> >
> > Besides, I accept the authors' argument that functional correctness is the primary bottleneck for FPGA-based prototyping and security applications. However, please ensure the final paper explicitly frames this "correctness-first" approach as a limitation for ASIC-focused workflows, where PPA is critical.
> >
> > Although achieving ~96% pass rates on VerilogEval-Human is significant, my concern, **about the core contribution being an engineering refinement (spec/prompt optimization) rather than a fundamental architectural shift**, is still remains.

---

### Official Review · Reviewer_19Jy · 2025-10-31

**Soundness:** 3
**Presentation:** 2
**Contribution:** 3
**Rating:** 6
**Confidence:** 4

**Summary:**

This paper focuses on the crucial but overlooked role of natural language descriptions/specifications (NLDs/specs) in LLM-based register-transfer-level (RTL) hardware design automation. It points out systematic shortcomings in current benchmarks, where low-quality, ambiguous, or over-detailed task descriptions suppress LLM performance and induce false negatives in code validation. Through extensive analysis and case studies, the authors develop an LLM-based refined spec creation pipeline (including evaluator, converter, and creator tools) that produces clearer, appropriately scoped specs, empirically enabling large general-purpose LLMs (e.g., GPT-4) to achieve markedly higher pass@5 rates on standard Verilog code generation benchmarks without additional fine-tuning or self-fixing. The work further demonstrates that this approach improves LLM performance even on more complex design tasks and provides actionable tools for dataset creators and engineers.

**Strengths:**

1. The motivation of this paper (specification failures) is insightful and is analyzed with concrete case studies (Section 3.1/3.2);
2. A systematic methodology for specification refinement backed by the development of practical LLM-powered evaluator, converter, and creator tools is proposed;
3. Convincing experimental results is provided: general-purpose LLMs like GPT-4, provided only with high-quality specs (not fine-tuned on hardware data or reliant on expensive post-generation self-fixing), can achieve comparable performance to existing works.

**Weaknesses:**

1. According to Table 1, only Qwen-Coder and GPT-4 were tested under the new NLDs. I suggest that the authors evaluate this specification refinement methodology on additional general-purpose LLMs and specialized RTL generation models to demonstrate its generalizability;
2. In the Abstract, the authors claim that “this low accuracy is affected by using low-quality descriptions as prompts in both training datasets and benchmarks.” However, the paper provides no discussion on the issues related to training datasets. Furthermore, applying the proposed specification refinement method to these training datasets (1) poses greater challenges, as the RTL code therein is often fragmentary or erroneous and does not constitute a golden model (GM), and (2) may not intuitively yield benefits for enhancing model performance.

**Questions:**

Please refer to the Weaknesses section.

---

> ### Author Response · Authors · 2025-11-13
> **Author response for reviewer 19Jy**
>
> Dear reviewer 19Jy,
>
> Thank you for your comprehensive comments and suggestions. While extended experiments are underway, we hereby provide some clarifications that may be of interest to you.
>
> **For weakness 1:** We will present more experiment results on cross generation: use one model to create the spec, and another model to generate code. We expect this can show the generalizability, and also prevent the model from "embedding" any information, which only itself can understand, into the spec.
>
> **For weakness 2:** Unfortunately, we cannot carry out finetuning experiments now due to the limited time. However, in the training dataset, the mismatch between the NLDs and the GMs can be regarded as a sort of mislabeling, and will theoretically downgrade the accuracy of the trained model. This is why we believe that refining the training dataset is necessary despite the lack of experiments.

---

### Author Response · Authors · 2025-11-13

Thank you for your comprehensive comments and suggestions. A revision of the paper, according to these suggestions, will be submitted after we finish the generalizability experiments, introducing additional models for both spec and code generation. While this may take some time, we can answer some of the questions based on existing results or theoretical considerations. We will soon comment on the reviews individually.

---

### Author Response · Authors · 2025-11-26
**Summary of Rebuttal**

Dear PCs, SACs, ACs, Reviewers, and all,

Thank you for the time reviewing our paper and the valuable comments and suggestions. To our best effort, we addresed the weaknesses and answered the questions in the latest revision of our paper. To help you and future readers better understand our work, we summarize our rebuttal and updates as follows.

----

**Common concerns**

- **On the generalizability and robustness of the specs**: More models are used for both code and spec generation, and we applied cross-evaluation, use one model to generate code based on specs created by another model, to prevent "embeddings" in the spec. The additional experiments show that the improvements brought by our specs consists in most configurations.

- **On the novelty of the work**: Despite the simplicity of our methodology, our novelty lies in a statistical diagnosis of dataset deficiencies, and an actionable spec creation framework that brings validated correctness gains across models and benchmarks. As we observed that the deficiencies we mentioned are appearining in outstanding works, we think that a foundational step is needed for future dataset and benchmark designers.

- **On the details of the tools and reproducability**: The prompts used by the tools are listed in the appendix, and the results can be reproduced with LLMs that are publicly available. We plan to open our repositories, containing the workflow code and the generated specs and RTL codes , as soon as possible.

----

**Individual concerns**

- **For reviewer 19Jy**: We analyzed the quality of exisiting datasets to show how does the dataset deficiencies affect the quality of the trained model. We leave as future work to train models using the refined specs.
- **For reviewer 9Vxd**: We analyzed the failing cases in the RTLLM benchmark to show to what extent can refined specs can fix the FNs. We also stated the reason why current LLMs are not yet ready for PPA optimizations, and why low PPA designs are still useful for accelerators and security-aiming designs.
- **For reviewer 3WLA**: We rewrote the "Spec design" section to give a more clarified definition between essential and detrimental features. Our studies on exisiting datasets shows the difference between our spec and existing works. Also, we added ethics, reproducability, and LLM usage sections.
- **For reviewer mB8U**: We confirmed that the spec we required is not in the training data.

----

We sincerely thank you again for your time and suggestions, and we are looking forward to the further feedbacks.

---

### Meta-Review · Area_Chair_CXDg · 2026-01-06

**Summary:**

This paper looks at the problem of using LLMs to gernreate RTL hardware designs, and located a problem of low performance of prevous stasks is because of the the use of low-quality descriptions as prompts in both training and benchmarks. They then designed an LLM-based spec refinement pipeline and helped the system to get a higher task on the RTL benchmarks.

**Reviewer Concerns:**

- The system is more on engineering an agentic flow rather than contributions on the scientific side
- Limited evaluation scope (partially addressed)

**Reviewer Scores:**

The author's summary actually indicated how the scores would change

Reviewer 19Jy: 6 -> (not yet replied) Reviewer 9Vxd: 2 -> 4 Reviewer 3WLA: 4 -> (not yet replied) Reviewer mB8U: 4 -> 4

In theory, this breaches fairness by pushing the boundaries. However, one could also arrive at this conclusion from examining the review.

To this end, even with these score adaptions, the paper is on boarderline rejection.

---

### Decision · Program_Chairs · 2026-01-26

Reject